# High-Impact Polystyrene Structured Light Components for Terahertz Imaging Applications

**DOI:** 10.3390/s25010131

**Published:** 2024-12-28

**Authors:** Kasparas Stanaitis, Vladislovas Čižas, Augustė Bielevičiūtė, Ignas Grigelionis, Linas Minkevičius

**Affiliations:** 1Department of Optoelectronics, Center for Physical Sciences and Technology (FTMC), Saulėtekio Ave. 3, LT-10257 Vilnius, Lithuania; vladislovas.cizas@ftmc.lt (V.Č.); auguste.bieleviciute@ftmc.lt (A.B.); ignas.grigelionis@ftmc.lt (I.G.); linas.minkevicius@ftmc.lt (L.M.); 2Institute of Photonics and Nanotechnology, Vilnius University, Saulėtekio Ave. 3, LT-10257 Vilnius, Lithuania

**Keywords:** HIPS, 3D printing, terahertz, terahertz beam forming, THz imaging

## Abstract

Terahertz frequency range imaging has become more and more attractive for a wide range of practical applications; however, further component optimization is still required. The presented research introduces 3D-printed high-impact polystyrene (HIPS) beam-shaping components for the terahertz range. Gaussian, Bessel, and Airy beam-shaping structures are fabricated, and different combinations are employed to evaluate imaging system performance. The combination of the Gaussian element as focusing and the Bessel element as collecting is revealed to be similarly efficient and less sensitive to misalignment than the classical Gaussian–Gaussian element setup. The presented research paves the way for introducing cost-effective structured light beam-shaping elements into THz imaging systems.

## 1. Introduction

The terahertz (THz) frequency range has recently become an attractive platform for next-generation electronic devices [1,2,3]. Located between the infrared and microwave wavelengths, it bridges optics and electronics, supplementing the advantages of both fields [4]. The synergy is evident in the growing number of proposed and/or realized practical applications. These include astronomy [5,6], medicine [7,8], pharmacy [9], material quality control [10,11], security [12,13], communication [14,15,16], and imaging [17,18], to name a few.

To speed up the transfer from research to real-life applications, the simplicity and cost-efficiency of the final solution are some of the most limiting factors. In most applications, the typical system comprises at least one emitter/detector pair [19,20] and the beam-shaping elements [21]. Since much effort has been devoted to the research and optimization of the THz frequency range detectors [22,23] and emitters [24], our article contributes to the less-addressed field of increasing the simplicity and price efficiency of the beam-shaping elements operating in the THz/sub-THz frequency range.

Before delving into the classical refraction-based beam-shaping components, the growing interest in metamaterial-based structures for the THz range is worth mentioning. The possibility of the metasurfaces changing the incident radiation in the metacell geometric parameter-dependent way was noted considerably a long time ago [25,26]. One of the most employed metacell shapes—the split ring resonator—operates due to the electric- or magnetic field-induced currents, resulting in a dipole or quadrupole formation in the structure with the controlled phase shift [27]. Thus, proper optimization of the metacells, comprising a metasurface, may achieve many different beam-shaping effects. For example, the arrangement according to Fresnel rules results in the formation of focusing Gaussian lenses [28]. The proposed structure was shown to be possible for polarization manipulation achieved via rotation. However, such designs are highly resonant, which limits their versatility. To address this, additional measures can be implemented to enhance beam-shaping control. One such measure is shown in reflectarray antennas, wherein the metacells, comprising a metasurface, are additionally featured by micro-electromechanical system (MEMS) switches [29] or mechanical manipulation [30], supplementing beam-front and spectral control for a single metasurface. Finally, even more enhanced beam-shaping possibilities may be achieved with more complicated structures. For example, it is shown that vanadium dioxide (VO_2_) may be employed in switchable chiral mirrors, wherein asymmetry of the reflection appears for the left- and right-handed polarizations (LCP/RCP) due to the heating and subsequent insulator-to-metal transition of vanadium oxide [31]. Another similar effect-employing approach has been revealed in double-layered SRR metasurface structures, wherein a dielectric layer is placed between two conductive metasurface layers. Such an approach supplements control of optical activity and refraction of the metamaterial switching between chiral and achiral resonator geometries [32]. Finally, employment of the double metasurface structures is shown to convert the incident radiation of linear polarization into circularly polarized waves. Employment of vanadium dioxide supplements thermal switching of asymmetric transition [33].

Traditional fabrication of beam-shaping structures employs materials such as z-cut quartz [34], silicon [35,36], metals [37], etc., which require specialized manufacturing processes, limiting their practicality and significantly increasing their price. Further improvement results in the need for alternative methods and materials that are inexpensive and easy to fabricate while still offering top performance. The recently proposed solution to employ paraffin as a cheap alternative has already been shown to be capable of providing an imaging quality comparable to that of classical solutions [38,39]. The disadvantage of such an approach is that paraffin is prone to mechanical stress and is unstable at temperatures above 60 °C. On the other hand, high-impact polystyrene (HIPS) has recently emerged as a strong candidate for easy and fast THz lens manufacturing, offering both affordability and performance [40]. Spectral analysis reveals high HIPS transmission in the frequency range from 0.1 to 1.3 THz [41], allowing it to be used for the fabrication of beam-shaping components [38,42]. Since HIPS’ melting temperature is typically around 200 °C, the optical components can be easily manufactured employing the widely available 3D printing technology. Such an approach allows for easy on-demand customization: fabrication of differently shaped components has already been revealed, including simple refractive lenses [39], parabolic reflectors [43], polarizers [44], and diffractive optics [45].

Further optimization of the imaging systems is expected by the employment of structured non-paraxial beam-shaping components [46]. The term “structured light”, currently experiencing a peak of scientific interest, unifies many optical fields, wherein inhomogeneities of field parameters are observed [47]. Recently, such elements have piqued interest for novel THz imaging system designs because of their unique advantages [48,49]; for example, the Bessel beam is non-diffractive, allowing for extended beam focus [50], and the Airy beam follows a curved trajectory and has self-healing properties [51]. The introduction of such beam-shaping elements has already revealed advanced imaging possibilities [17,36]. However, previous research employed flat optical components that are considerably complex to manufacture, despite their advantages. Furthermore, the research mentioned above reveals imaging analysis introducing a single non-paraxial element placed before the sample to be imaged. Thus, there is a strong interest in studying the imaging capabilities of the THz system employing two non-paraxial beam-shaping components. Their positioning as the focusing and collecting lenses create unique beams that must be measured experimentally to evaluate which non-paraxial lens pair provides the best imaging quality.

In this research, enhancement of the THz imaging system provided by the insertion of non-paraxial HIPS-based beam-shaping components is explored. The paper presents three elements that shape the incoming electric field to Gaussian, Bessel, or Airy distribution, which have been designed and printed using a conventional 3D printer. The properties of the fabricated HIPS lenses are revealed by the time-domain spectroscopy (TDS) and beam characterization setup. The performance is further studied and compared by employing the single-pixel imaging setup with different combinations of the produced beam-shaping elements by imaging a USAF 1951 sample modified for the THz frequency range. System sensitivity to sample misalignment is also analyzed for the best revealed beam-shaping structure configurations.

## 2. Materials and Methods

In this study, extrusion 3D printing was employed to manufacture optical components. The approach has already been validated for the fabrication of the beam-shaping elements, operating in the THz frequency range [52,53]. The fabrication method is also favored due to its wide availability, robustness, and cost-efficiency. In the sub-THz domain, large wavelengths of interest reduce concerns about diffraction from infill patterns [54,55]. The optical components were printed using a Creality K1 3D printer (Creality 3D Technology Co., Ltd., Shenzhen, Guangdong, China), with a 400 μm nozzle diameter and a 100 μm vertical step size. A 100% infill density was maintained to ensure a constant bulk refractive index, crucial for optimal optical performance. It is worth noting that there is a considerably wide range of polymers exhibiting high transmission and suitable refractive indices in the frequency range of interest, making them ideal for fabricating optical components [56]. For this particular research, high-impact polystyrene (HIPS) was selected [38,39]. The choice for this particular fabrication material is based heavily on the fact that HIPS is one of few 3D extrusion printer-compatible materials where the absorption coefficient does not increase as rapidly with increasing frequency; only polypropylene (PP) shows a better absorption coefficient at higher frequencies [42], but are considerably more difficult to print due to wrapping [57,58]. THz time domain spectroscopy (THz-TDS) reveals a refractive index of 1.53 at 253 GHz, as depicted in Figure 1a.

Three types of optical components were fabricated: a hemispherical Gaussian lens, a $15^{\circ }$ axicon lens generating a Bessel beam, and a structure corresponding to a third-order 2D function, resulting in Airy phase distribution (see Figure 1c). The shape of the hemispherical Gaussian lens is characterized by the curvature (R) and lens thickness (D). The parameters were optimized employing finite difference time domain (FDTD) method simulations, aiming for the focal distance of 30 mm (from the top of the structure). The parameters were found to be R=27mm and D=9mm. The axicon lens is defined by a single cone angle (α) parameter, which corresponds to the base angle of the triangular cross-section. An angle as small as $\alpha =15^{\circ }$ was selected to achieve a high depth of focus (DOF). The Airy lens phase profile was calculated as a third-order two-dimensional function [17]:(1)$$\Phi_{\mathrm{AI}}\left(x,y\right)=S\left(x^{3}-y^{3}\right),$$
where $S=3.428\cdot 10^{5}$ m^−3^ is the scaling coefficient, defining the slope of the function and ensuring the possibility of manufacturing the structure with the 3D printing technique. The fabricated components are presented in Figure 1b. The performance of the designed structures has been checked using the finite difference time domain (FDTD) method. Figure 1c reveals the X−Z cross-section of the corresponding lens.

Fabricated beam-shaping structures were characterized to assess their focusing capabilities and spatial distributions. A beam-shaping setup (see Figure 2a) was employed for this purpose. The THz radiation at 253 GHz was generated using a 65 nm CMOS differential Colpitts oscillator, which leverages capacitive and inductive feedback loops to achieve stable fundamental oscillations near 84 GHz, and via enhancement of the third-harmonic components, delivers a power output of 78 µW at 253 GHz [59]. Beam-shaping characterization was performed using a narrow band 300 GHz microbolometer detector mounted on 3D motorized linear stages. Such a detector employs an air-bridged Ti microbolometer coupled with a resonant dipole THz antenna, wherein the resonant frequency is defined by the geometrical parameters of the antenna [60]. Additionally, the components were tested in a THz single-pixel imaging setup (see Figure 2b), wherein the detector was fixed and the sample was mounted on the motorized stages. It is worth noting that a different broadband log-spiral antenna coupled with a nanometric FET detector, based on a 90 nm CMOS technology, was employed [61], which is more sensitive, and thus higher imaging contrast can be recorded. Such a detector was not employed for the beam-shaping characterization, as it comprises a top-mounted lens, featuring a larger signal collection area, hindering the scanning process but not impacting the single-pixel imaging measurements as the collecting lens (marked $L_{2}$ on Figure 2) diminishes the enlarged collection area effect. A band-pass filter, manufactured from 30 µm steel foil with specifically size-optimized rectangular holes, allowing the transmission of only a narrow band of frequencies in a certain polarization, was inserted before the detector [62]. The filter cells were spaced 780 µm apart both horizontally and vertically and consist of a rectangular cut-out, with the length being 624 µm and the width 92 µm. The central wavelength of the resulting filter was at 244 GHz, with a bandwidth of 36 GHz, and the central wavelength shifted to the lower frequency side due to manufacturing imperfections. Lens characterization raster scanning was performed using the step size of 0.25 mm, while imaging was performed at 0.15 mm, since a higher resolution was required to evaluate the imaging performance accurately. The imaging setup was employed to evaluate the performance of the imaging system comprising seven different combinations of focusing-collecting lenses: every possible pair of the three fabricated lenses and a conventional Gaussian–Gaussian lens combination as a reference.

A USAF 1951 target, modified for the THz frequency range, was imaged. The target was fabricated using laser ablation on 30 µm steel foil. Ablation was performed using a Pharos SP laser (Light Conversion, UAB, Vilnius, Lithuania) employing ultrashort pulses in the 0.158 ps–10 ps range, with the central wavelength λ = 1030 nm. The average power reaches 6 W at a repetition rate of 200 kHz. Employment of such a target allows the acquisition of classical imaging system characterization benchmarks like the modulation transfer function (MTF), evaluated using different sizes of periodic slits on the sample, and the mean square error (MSE) image comparison method.

## 3. Results and Discussion

### 3.1. Beam Profiling

The beam-shaping abilities of each structure were evaluated by 2D scanning along the optical axis and at the focal plane. The results are presented in Figure 3. Color reveals the intensity normalized to the average incident non-focused (collimated) beam intensity. The full-width at half-maximum (FWHM) values for the Gaussian, Bessel, and Airy lenses were 1.8 mm, 2.6 mm, and 3.0 mm on the first lobe, respectively. Additionally, depth of focus (DOF) was evaluated for each lens. For the case of the Gaussian lens, depth of focus is expressed as DOF $=2Z_{R}=23$ mm [63], wherein $Z_{R}$ is the Rayleigh length. The DOF of the Bessel beam structure was evaluated by using a simplified equation
(2)$$\mathrm{DOF}=\frac{r\sqrt{1-n^{2}\sin^{2}\alpha }}{\sin\alpha \cos\alpha \left(n\cos\alpha -\sqrt{1-n^{2}\sin^{2}\alpha }\right)}\approx \frac{r}{(n-1)\alpha },$$
where *r*—radius of incident radiation, *n*—refractive index, and α—axicon characteristic angle. The DOF of the Bessel structure was found to be DOF =108 mm [64]. The DOF of the central lobe of the Airy structure was found to be DOF =33 mm [65]. The acquired results indicate that, while the Gaussian lens achieves the smallest focal spot, it has a short focal depth. In contrast, the Bessel and Airy lenses exhibit a higher DOF but a less-concentrated focal spot. Thus, the employment of the Gaussian structure is expected to deliver better results. However, its calibration is more complicated than the other two beam-shaping structures due to the small DOF. Bessel and Airy structures, on the other side, despite having smaller radiation concentration abilities, are much easier to align due to a significantly larger DOF.

### 3.2. Imaging

The single-pixel imaging was performed for the seven lens combinations: GG, GA, GB, BG, BA, AG, and AB. Letters correspond to the employed structures marked $L_{1}$ and $L_{2}$, respectively (see Figure 2b). *G* corresponds to Gaussian beam structure, *A*—Airy, and *B*—Bessel. The resulting imaging and the corresponding mean square error (MSE) image comparison values are presented in Figure 4 (the acquired images are compared with the binary image, comprising slits of the same size). A clear variation in performance is observed, depending on the combination of lenses.

First, assuming the system is aligned ideally, the best imaging result is expected upon the classical solution, employing two Gaussian structures. In such a configuration, the largest part of the incident radiation is focused in and collected from a considerably smaller point than in other solutions. On the other hand, the smallest DOF of the Gaussian lens makes such a combination most sensitive to improper setup alignment. Authors do believe that imaging employing a classical imaging setup comprising two Gaussian structures (GG) should deliver the lowest MSE of the compared configurations in ideal investigation conditions; thus, the MSE of the GG not being the smallest one is most likely explained by minor misalignment, which occurred during the experiment. The revelation of the problem upon experimental conditions also reveals the obvious troubles expected to appear in practical applications, especially in focal point adjustable systems. Thus, the employment of different beam-shaping structures, as presented in this work, may serve as a solution to simplify system calibration, especially useful for practical implementation. One may note that the lowest MSE (corresponding to the best image quality) was achieved for the Gauss–Bessel imaging system.

Furthermore, one may note the non-interchangeability of the focusing and collecting lenses. This is most obviously visible when comparing GB/BG and GA/AG cases. The MSE of the systems, wherein the Gauss beam is employed as a focusing element and the beam of the wider DOF is used as a recollecting element, is considerably smaller. Employment of the wide DOF element as a collecting lens allows the diminishment of the misalignment error, and the employment of the Gaussian lens as a focusing element concentrates radiation, thus allowing it to collect the most significant portion of the signal close to the sampling plane. The importance of the proper signal concentration on the sampling plane becomes obvious when analyzing images acquired by the beam-shaping element combinations without a Gaussian lens (BA and AB). As one may note, both combinations facilitate the least appealing imaging results, leading to the naked-eye visible distorted image. Still, it is essential to note that only the binary sample (only full transmission or full reflection are expected during imaging) was analyzed during the presented research. An employment of samples comprising transmissive materials may yield different results, leading to different, more optimal lens combinations.

The above-described results were additionally supplemented by the modulation transfer function (MTF) dependencies, revealed in Figure 5. The dependencies were acquired using the above-presented images. One may note that the highest contrast is revealed for the GG and GB configurations. To further analyze the reliability of the investigated systems, the sample-defocusing effect on the imaging quality in the GG and GB configurations was analyzed. The focusing and collecting lenses were fixed and the X−Y−Z imaging of the THz frequency range-modified USAF1951 sample was performed with a constant Δz=2.5 mm shift, acquiring MSE values of the each image. Such a setup is expected to represent one of the possible imaging setup misalignments, wherein the sample to be imaged is placed out of focus of the system. The graph in Figure 6 reveals the MSE dependencies on the sample shift. Firstly, it is worth noticing that the MSE of the GG system is almost always higher, possibly due to the above-discussed reasons. What is considered much more interesting is that the relative change of the MSE (depicted in percent in the inset of Figure 6 for further clarity) is considerably lower for the GB system compared with the GG case. This result supports the above claim that the GB system is less sensitive to the component misalignment.

## 4. Conclusions

The presented study covers an important challenge of THz imaging technology transfer into the practical application field by exploring solutions to increase price-efficiency and reliability of the imaging systems. During the research, three types of beam-shaping elements were fabricated using the widely available 3D printing technology. Structured light elements forming Gaussian, Bessel, or Airy beams were fabricated and characterized. The designs of the structures were confirmed using finite difference time domain (FDTD) method simulations, the structures were fabricated using 3D printing, and their optical properties were investigated by time domain spectroscopy (TDS) and THz beam characterization setup. The fabricated structures displayed characteristics predicted by the simulation—the smallest full-width at half-maximum (FWHM) was found for the Gaussian beam. At the same time, the largest depth of focus (DOF) was observed with the Bessel beam-forming structure.

The mean square error (MSE) value was chosen as a benchmark to determine imaging setup performance. The best images (lowest MSE) were achieved using the Gauss–Bessel (GB) system, wherein the Gauss element was radiation focusing and the Bessel element was radiation collecting. The effect is explained by the large DOF of the Bessel structure, leading to less strict setup alignment requirements. The Gauss–Gauss (GG) system revealed similar imaging results. Additionally, the defocusing effect on the system performance was analyzed by sweeping the sample to be imaged along the optical axis between two lenses. Considerably smaller MSE changes were recorded for the GB system compared with the GG-based imaging setup. This result reveals that the GB system is more prone to sample misalignment errors than the GG imaging system.

Structured light-forming elements, found to be less effective for the presented research (like imaging setups with the Airy beam-forming structure or setups with the Bessel beam-forming structure as a focusing element) are expected to contribute beyond the limits of a classical Gaussian system in imaging specific targets. Thus, a double Bessel lens with a specifically optimized depth of focus may be preferential for imaging thick samples, and an Airy lens may be employed for imaging around opaque obstacles, employing the self-healing ability of the Airy beam. Further integration of the proposed structures with similarly cost-effective metasurface-based beam-shaping solutions is also expected to significantly increase structure performance and efficiency of the THz frequency range-based systems.

## Figures and Tables

**Figure 1 sensors-25-00131-f001:**
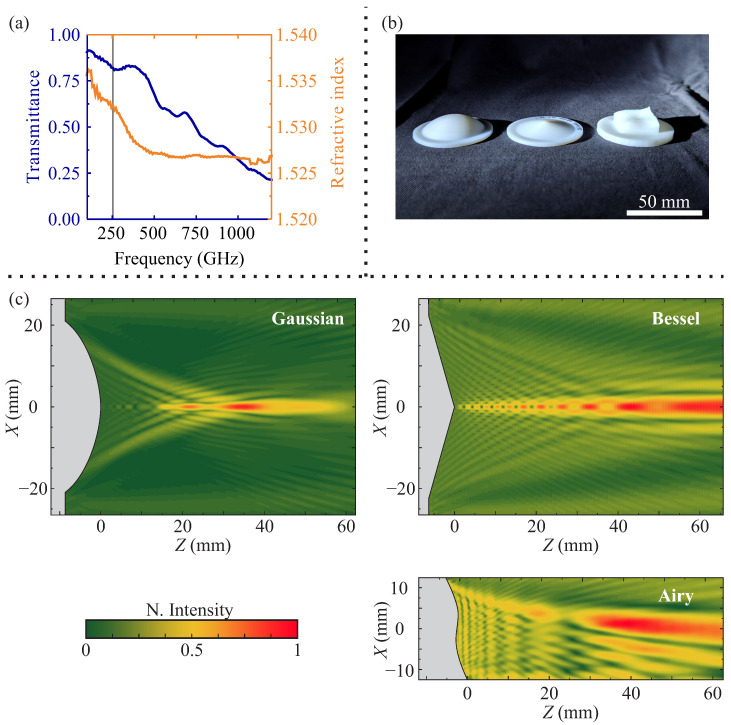
(**a**) Spectral dependencies of transmittance and refractive index of the employed high-impact polystyrene (HIPS). The gray vertical line depicts the position of the 253 GHz, corresponding to the frequency of the resonant source, employed within the presented research. (**b**) Photo of the 3D-printed lenses employed within the presented research; here, the diameter of each component is equal to 50.8 mm. (**c**) Simulation, revealing electric field distribution along the optical axis (X−Z plane) for the fabricated structures. Simulations were conducted using the finite difference time domain (FDTD) method. Note the appearance of the focused point for the case of the Gaussian lens, the longitudinal focused beam for the case of the Bessel lens, and the curved longitudinal focused beam for the case of the Airy lens.

**Figure 2 sensors-25-00131-f002:**
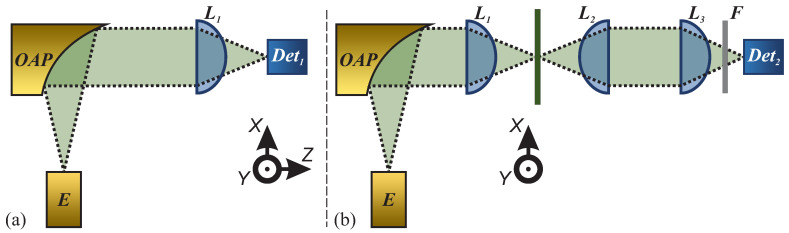
Sketch depicting setups employed during the presented research. (**a**) Beam characterization setup: THz radiation source, resonant at 253 GHz (E), collimated by off-axis parabolic mirrors (OAP), is incident on the beam-shaping element to be characterized $\left(L_{1}\right)$. The detection is performed by the microbolometer detector, resonant at 300 GHz $\left(Det_{1}\right)$, placed on the 3D linear stage system, allowing the acquisition of a 3D image of electric field distribution beyond the analyzed element. (**b**) Sketch depicting the classical single-pixel imaging setup: the first beam-shaping element $\left(L_{1}\right)$ is expected to focus radiation on the target to be imaged, and the focused radiation is collected by the second element $\left(L_{2}\right)$. The third beam-shaping element focuses radiation into the detector. The LogSpiral detector $\left(Det_{2}\right)$, supplemented by the polarizing filter (F), was employed for the single-pixel imaging. The imaging is performed by scanning via moving the target to be imaged.

**Figure 3 sensors-25-00131-f003:**
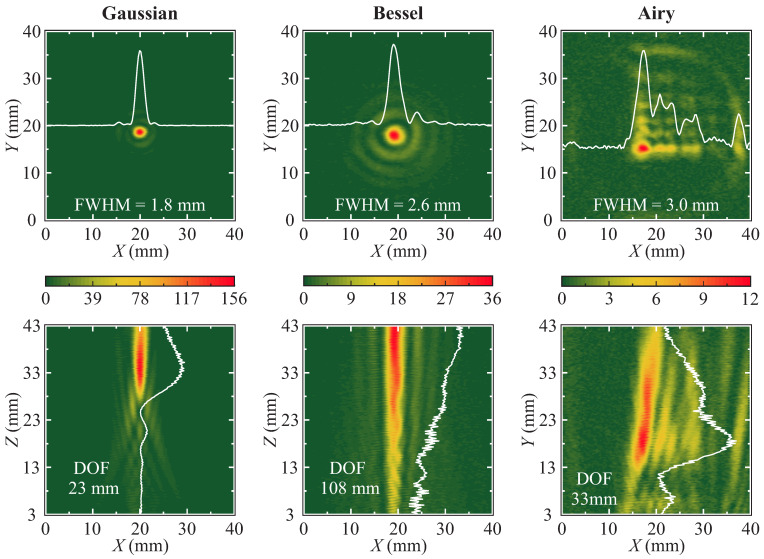
Sketch depicting the beam-shaping abilities of different fabricated structures, revealed via 2D scanning of focal plane (X−Y, **upper row**) and along the optical axis (X−Z, **bottom row**). Note the smallest FWHM of the Gaussian beam and the largest depth of focus for the Bessel beam. White lines represent intensity profile through the center of the focal plane (**top row**) and along the optical axis (**bottom row**).

**Figure 4 sensors-25-00131-f004:**
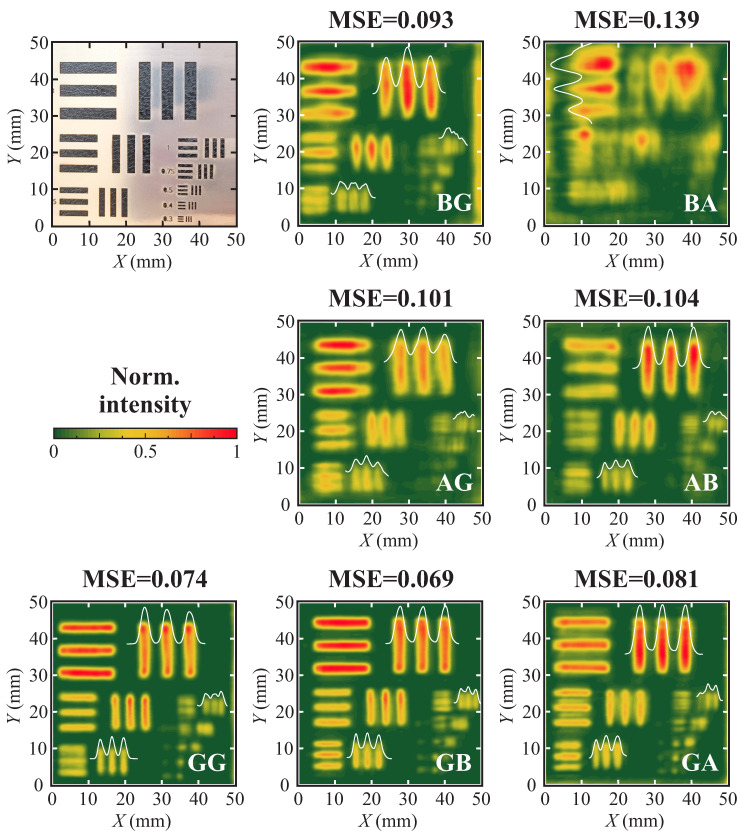
Single-pixel images of the modified USAF1951 target (**top left image**) performed employing different combinations of focusing-collecting beam-shaping elements. *G*—Gaussian beam structure, *B*—Bessel beam, and *A*—Airy beam. Note the difference of the imaging quality expressed via the mean square error (MSE) benchmark.

**Figure 5 sensors-25-00131-f005:**
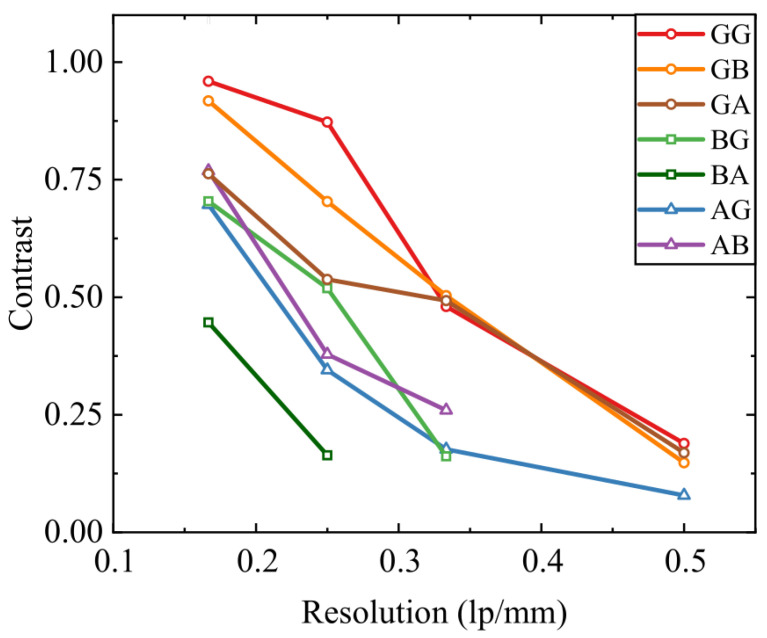
Modular transfer function (MTF) dependency for the above-described single-pixel imaging system, comprising different radiation focusing and collecting element combinations. MTF was acquired by imaging the USAF1951, modified for the THz frequency range. *G*—Gaussian beam structure, *B*—Bessel beam, and *A*—Airy beam.

**Figure 6 sensors-25-00131-f006:**
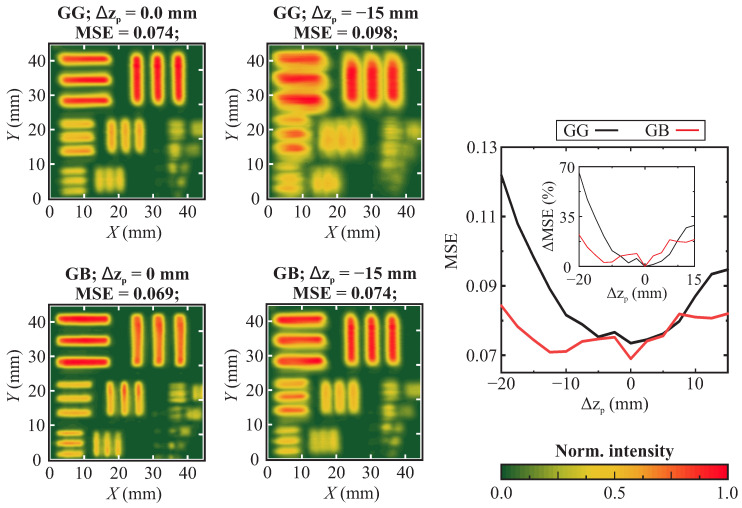
Sample defocusing effect on the image quality in Gaussian–Gaussian (GG) and Gaussian–Bessel (GB) imaging systems. Mean square error (MSE) is employed as a benchmark. Most left images depict the case of the lowest MSE (best image), and the images in the center depict images taken at Δz=15 mm sample shift. The performed scenario corresponds to the out-of-focus imaging. Right: MSE dependency on the sample shift. Inset: depicts MSE change dependency (in percent compared with the lowest achieved value) on sample shift. In both graphs, note the considerably lower MSE dependency on the sample shift for the case of the GB imaging system.

## Data Availability

The data presented in this study are available on request from the corresponding author.

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
