# Peer review of "High-Impact Polystyrene Structured Light Components for Terahertz Imaging Applications"

_sensors, 2024, doi:10.3390/s25010131_

Round 1
Reviewer 1 Report
Comments and Suggestions for Authors
3D printed polystyrene beam shaping components for THz applications are presented. The authors well described the research motivation of the components for THz ranges. The previous research involved with beam shaping components are well described. Therefore, I can understand the research trend about the shaping components used for THz imaging applications.
Figure quality looks very good. Figure sketch depiction for the presented research for beam components is understood. The single-pixel imaging for THz components is very clear to be observed. The MTF graphs for single-pixel imaging is also reasonable. The MSE graphs in the defocusing effect on the image quality in GG and GB imaging systems are very clear. In Conclusion section, the authors well summarize the characteristics of the beam shaping components for THz applications, motivation of the several THz components.
In addition, there are no English grammar problems at all. The English expression is smoothly well described because I can fully understand the proposed idea and description, It is very hard to find something wrong about the description, mathematical approaches, measurement, and conclusion. Therefore, the submited manuscript is fully accepted as it is because the submitted manuscript is a good example of the manuscript.
Author Response
We want to thank the reviewer for the positive feedback.
Reviewer 2 Report
Comments and Suggestions for Authors
The paper titled "High-Impact Polystyrene Structured Light Components for Terahertz Imaging Applications" by Kasparas Stanaitis et al., presents a study on the development and application of 3D-printed High-Impact Polystyrene (HIPS) components for terahertz (THz) imaging. Here is a review of the paper based on its content:
1. A deeper analysis of the material properties and their impact on the long-term stability and performance of the HIPS components would be valuable.
2. Comparing the performance of HIPS components with other materials or structures could provide a more comprehensive understanding of their advantages and disadvantages.
3. The paper could benefit from discussing potential avenues for further optimization of the beam-shaping structures, especially regarding their alignment and integration into existing THz imaging systems.
Reviewer 3 Report
Comments and Suggestions for Authors
In this manuscript, Kasparas Stanaitis and co-authors demonstrated three types of 3D-printed high-impact polystyrene beam-shaping components for the terahertz applications, including Gaussian, Bessel, and Airy beam-shaping devices. This study presents interesting results which would be of interest to researchers in the field of beam shaping, imaging and other related fields. However, revisions are required before the manuscript can be accepted by Sensors.
1. What is the efficiency of the proposed Gaussian, Bessel, and Airy beam-shaping devices?
2. It is recommended that the author indicate the specific device size in Figure 1 (b).
3. The Z in Figure 1 (c) should be italicized.
4. The authors emphasize that their proposed method is a cost-effective one. Can the author specify and compare how cost-effective this method is?
5. There are some recent papers about terahertz technology, which are suggested to be cited so as to set a more enriched reference background.
Advanced Functional Materials 31(14), 2010249, 2021. Advanced Optical Materials 8(15), 2000247, 2020. Scientific Reports 9(1), 4097, 2019.
